# Eating Healthy, Growing Healthy: Outcome Evaluation of the Nutrition Education Program Optimizing the Nutritional Value of Preschool Menus, Poland

**DOI:** 10.3390/nu11102438

**Published:** 2019-10-13

**Authors:** Joanna Myszkowska-Ryciak, Anna Harton

**Affiliations:** Department of Dietetics, Institute of Human Nutrition Sciences, Warsaw University of Life Sciences (WULS), 159C Nowoursynowska Str, 02-776 Warsaw, Poland; anna_harton@sggw.pl

**Keywords:** nutrients, dietary guidelines, nutrients density, diet quality, preschool menu, education

## Abstract

Staff education can improve the quality of nutrition in childcare centers, but an objective assessment of the change is necessary to assess its effectiveness. This study evaluated the effectiveness of the multicomponent educational program for improving the nutritional value of preschools menus in Poland measured by the change in nutrients content before (baseline) and 3–6 months after education (post-baseline). A sample of 10 daily menus and inventory reports reflecting foods and beverages served in 231 full-board government-sponsored preschools was analyzed twice: at baseline and post-baseline (in total 4620 inventory reports). The changes in 1. the supply of nutrients per 1 child per day; 2. the nutrient-to-energy ratio of menus; 3. the number of preschools serving menus consistent with the healthy diet recommendations, were assessed. Education resulted in favorable changes in the supply of energy, fat and saturated fatty acids. The nutrient-to-energy ratio for vitamins A, B_1_, B_2_, B_6_, C, folate and minerals Calcium, copper, iron, magnesium, phosphorus, potassium and zinc increased significantly. The percentage of preschools implementing the recommendations for energy, share of fat, saturated fatty acids and sucrose as well as calcium, iron and potassium increased significantly. However, no beneficial effects of education on the content of iodine, potassium, vitamin D and folate were observed. This study indicates the potentially beneficial effect of education in optimizing the quality of the menu in preschools. However, the magnitude of change is still not sufficient to meet the nutritional standards for deficient nutrients.

## 1. Introduction

The quality of menu offered to children in different type of day care centers (DCCs) is important for the correctness of their diet. However, in the case of full-day and full-board (providing two main meals and 1–2 snacks per day) preschools, it is a key determinant of the children’s overall diet quality [1]. In this type of institution in Poland, children spend up to 10 h a day five days per week and eat most of their meals. Therefore, it is recommended that nutrition in these institutions should cover at least 70% of children’s daily energy and nutrients reference intake [2]. On average, children consume from 50% to 100% of foods they are served in DCCs [3]. During the period of the present study, 80.7% of Polish children aged 3–6 attended various forms of preschool education, which accounts for 1,299,138 children. Of these, 982,024 children were offered full board meals while staying in care, which demonstrates the important role of preschools ensuring proper and healthy nutrition of children in Poland [4]. However, research internationally [5,6,7] and in Poland [8,9,10,11,12] report that the quality of nutrition in preschools is not always consistent with dietary recommendations and standards. Analyses of menus carried out in individual childcare institutions in Poland pointed to an inadequate content of calcium [8,9,10,11], vitamin C [9,10], vitamin D [8,11] and iron [10,11] compared to the recommendations, as well as oversupply of energy and fat [9].

One of the reasons for this situation in Poland may be the lack of specific legal regulations regarding the nutrition of children in preschools. Most importantly, there is no central legal regulation concerning energy and nutrients supply in preschool menus. Polish dietary reference intake standards defined by the National Institute of Food and Nutrition [13] are not mandatory for preschools. Currently, applicable legal regulations are very general and relate to the supply of selected product groups and culinary techniques. By law, children in preschools should receive daily at least two servings of dairy products, one serving of meat or a meat alternative, two servings of grains and fruit or vegetables in every meal. Additionally, preschoolers should be served minimum one serving of fish and not more than two fried meals per week [14]. Unfortunately, no serving/portion size has been defined for any of the product groups. There is also a limit of sugar added to beverages (maximum 10 g for 250 mL) prepared in preschools from the scratch (institutions can buy and serve commercial beverages with the higher sugar content).

Another reason for the observed irregularities in preschools menus may be the lack of qualified personnel. The Polish legislation does not require the employment of a dietitian or nutritionist for planning and supervising the menu in preschools. A person with a secondary education, preferably with a gastronomic or economic profile may be employed as an officer responsible for purchases of food products and often also for menu planning [15]. Planning a preschool board by a person with no specialized higher nutrition education and based on the very general law regulations may result in a lack of implementation of the nutrition recommendations in preschool menus. Earlier studies conducted in Poland in individual DCCs suggested insufficient nutritional knowledge of their staff [16,17]. These initial observations were confirmed in a representative group of Polish child caregivers (*n* = 892) in 2014 [18]. In this case, improving the quality of menus can be achieved through the education and practical training of institutions’ staff [19,20]. Unfortunately, such educational activities in Poland are usually of a local nature [21]. Often, educational programs focus on a single aspect of the diet, e.g., fruit and vegetable supply, rather than the overall quality of the menu provided by the institution [22]. Moreover, they do not include evaluation of the effectiveness of education or the evaluation is based on assessing the knowledge of staff, not the actual change of nutrition in preschool [23]. In some cases, the effectiveness of the program is assessed based on the opinions of the institution itself, which may not be objective and lead to response bias [24]. The institutions participating in our education program declared a reduction in the amount of salt in menus [24], which, however, is contradicted by the quantitative data presented in this article. This clearly demonstrates the need to assess the effectiveness of education on the basis of measurable parameters, not declarations of DCCs.

Research indicate that properly balanced nutrition in DCCs can be a part of the strategy for preventing diet-related diseases, such as obesity [25]. Ensuring the proper nutrition of children in DCC is a global problem, and one of the common limitations is the lack of staff training [26]. Recently, much attention has been paid to assessing the effectiveness and efficiency of various types of interventions aimed at improving the quality of children’s nutrition. There is some evidence that specific school food environment policies can improve targeted dietary behaviors [27]. Lunchbox interventions are effective in improving the packing of vegetables in children’s lunchboxes, however, the impact on children’s dietary intake requires more research [28]. There is some evidence that parent nutrition education is not effective in increasing fruit and vegetable consumption in children aged five years and younger [29]. Due to the difficulties in obtaining measurable improvement in the case of parental education, the rational choice is the education of the personnel responsible for feeding children in care facilities. Knowledge about the effectiveness of different kinds of educational activities in changing the quality of preschool menus can be used when designing nutritional interventions, not only in Poland but also internationally.

Therefore, the purpose of this study was to evaluate the effectiveness of the multicomponent educational program for improving the nutritional value of preschools menus in Poland, measured by the change in nutrients content and adherence to nutritional recommendations before and after education.

## 2. Materials and Methods

### 2.1. General Information

The presented data were obtained in the framework of the program Eating Healthy, Growing Healthy granted by Danone Ecosystem and conducted within the years 2014–2017 in Poland. The main goal of the program was to improve the quality of nutrition in DCCs through a multicomponent nutrition education (including education materials, face-to-face educational meetings and/or audit and feedback) aimed at the staff responsible for the implementation of nutrition in these institutions. The program was nationwide; participation was voluntary and free of charge for participating institutions. The contact details of DCCs were obtained from the local relevant municipal institutions. All registered DCCs (nurseries, preschools) received an e-mail invitation to participate; in addition, information about the program was advertised on media channels addressed to care and educational centers (press, internet portals, etc.). To participate in the program, the institutions had to register on a dedicated website (http://zdrowojemy.info/placowka/zapisz-placowke-do-programu (in Polish)) until the end of the recruitment period.

When registering for the program, the institutions were offered a choice between two options: 1. A direct participation, including audit of the menu (10 consecutive days) and feedback at two time points (baseline and post-baseline), a 24-h educational training (lectures, workshops, counselling/consultation sessions) for personnel involved in ordering food products, planning and preparing menus in DCC, and ongoing support from a dedicated and specially trained educator; 2. an indirect participation, offering access to training materials (in form of Power Point presentations, practical exercises with solutions, handouts) on the Eating Healthy, Growing Healthy website platform (but no nutritional assessment of the menus and supervision of a dedicated educator). The detailed project description is available elsewhere [11,30], the overall scheme of the program with the thematic scope of educational materials are presented in the Appendix A.

The evaluation of the program was conducted at three levels: 1. an assessment by the educator, 2. an assessment by the DCC, 3. an objective assessment of the quality of nutrition in the DCC after education (effect of education) by a supervisor. The first two evaluations were conducted using an evaluation questionnaire (including questions about the level of satisfaction with participation in the program, the type of changes introduced in nutrition, the level of difficulty in implementing recommendations) among educators and randomly selected institutions [24,31]. The effectiveness of the program on the quality of nutrition in DCCs was evaluated on the basis of nutritional analysis of the DCC menu before (baseline) and 3–6 months after the education training (post-baseline) by the program supervisors. Such a period is used in this type of interventions [32]; it allowed institutions to change their food suppliers if necessary while not allowing the implementation of another educational program. Depending on the type of documentation provided by the DCC (menu or inventory reports), a qualitative assessment (menu analysis: culinary techniques, the presence of core food groups in the menu, the type of beverages served) or a quantitative assessment (inventory reports analysis: the supply of food products, energy and nutrients per child per day) was carried out. For technical and organizational reasons, the above-described assessments of the effectiveness of education were only possible for the directly participating DCCs.

In total, 2638 daycare institutions for children aged 0.5–6 years old (nurseries, preschools) from 16 voivodships in Poland were enrolled in the program and 13,214 employees were covered by the direct education. In this study, only directly participating preschools with completed education training and the quantitative nutritional assessment before and after education were included.

### 2.2. Ethical Approval

The program did not require the consent of the ethics committee as no research activities were deemed to be a human subjects research according to University of Life Sciences Center Institutional Review Board guidelines. No personal data concerning children attending DCCs and DCCs staff were collected within the program. Applicants (DCCs) were informed about the purpose and scope of the program, and the possibility of withdrawing from it at any stage without giving any reason with no consequences. Registering the institution on the dedicated website portal was tantamount to agreeing to participate in the program and acceptance of its regulations.

### 2.3. Study Participants and Design

The present study focused on the effectiveness of personnel education on the supply of energy and nutrients in the preschools’ menus. To ensure a homogeneous group for the analysis, the exclusion criteria described below were applied (Figure 1).

The private (non-public) DCCs were excluded due to previously reported [33] significantly higher average financial rate per child per day (8.3 ± 2.1 vs. 5.8 ± 1.3 PLN). A higher budget may have a potential impact on purchasing capacity, especially for the more expensive products (e.g., meat, fish). Finally, the following were qualified for the analysis: Government-sponsored (municipal/public) institutions offering full-board (main meals: breakfast, lunch, morning and/or afternoon snack), maintaining kitchen facilities and preparing all meals from scratch, providing full dietary records including inventory reports with all the food products used in the kitchen to prepare meals and the number of children eating on the day at the baseline and post-baseline (3–6 months after education). The study design is presented in Figure 2.

In total, 231 institutions from all 16 voivodships in Poland were qualified; 462 decade menus, and 4620 daily inventory reports were analyzed.

### 2.4. Nutritional Analyses of Preschools Menus

This study presents a comparison of quantitative analyses of preschools menu at the baseline and after the education (post-baseline). The presented data relate to 1. the supply of energy and nutrients per child per day in all preschools, 2. the nutritional density of menus in all preschools, 3. the number/percentage of preschools that meet the key recommendations for proper nutrition of children, before and after education.

The content of energy and nutrients was calculated on the basis of inventory reports and menus from 10 consecutive days (two preschool weeks). In Poland, menus for mass catering (care institutions, hospitals, etc.) are planned for a longer period (a week or ten days), which ensures a better balance in nutrient content and provides suitable diversity [34]. Inventory reports included all food products along with their quantity (in kilograms/liters/pieces) used in the kitchen to prepare all meals for children, along with the number of children who consumed these meals that day. This enabled an accurate calculation of quantities of products used to prepare meals for one child. The detailed procedure for calculating the nutritional value of preschool menus per child per day based on the inventory reports along with an example has been described elsewhere [11,30]. The dates were computed with software Energia v 4.1 (Copyright 1997/2006 by Andrzej Miegoć, Warsaw, Poland) with the Polish Nutrition Database [35]. In total, a sample of 4620 records on food and beverages served to children attending 231 preschools was analyzed. The average energy, macronutrients (total fat, saturated fatty acids, monounsaturated fatty acids, polyunsaturated fatty acids, cholesterol, total protein, animal/vegetable origin protein, total carbohydrates, sucrose, lactose, starch and dietary fiber), vitamins (A, retinol, beta-carotene, B_1_, B_2_, B_6_, B_12_, C, D, E, folate, niacin), and minerals (calcium, copper, iodine, iron, magnesium, phosphorus, potassium, sodium and zinc) content of the daily menu were calculated for one child.

The content of nutrients was referred to Polish dietary reference intake (DRI) [13]: (1) EER for energy (kcal); (2) RDA for protein (g), carbohydrate (g), calcium (mg), copper (mg), iron (mg), magnesium (mg), phosphorus (mg), zinc (mg), vitamin A (μg, as retinol activity equivalents), vitamin B_1_ (mg), vitamin B_2_ (mg), vitamin B_6_ (mg), vitamin B_12_ (μg), vitamin C (mg), vitamin E (mg), folate (μg, as dietary folate equivalents), niacin (mg); (3) AI for sodium (mg), potassium (mg), iodine (μg), vitamin D (μg), and dietary fiber (g); (4) the acceptable macronutrient distribution range for protein, fat, and carbohydrates (as a percent of total kilocalories/energy). For the cholesterol and saturated fatty acids level, the general population recommendations for Polish were used. The DRI for energy and nutrients in Poland are formulated separately for children aged 1–3 and 4–6. The preschool period typically includes children aged 4–6 years, however, children from the age of three can be enrolled. Since children aged 4–6 constitute the majority in preschools, for practical reasons, the menu is planned to meet their nutritional needs. However, in our analyses, we included both age categories separately.

Due to the large variation in the energy value of preschool menus, the nutrient-to-energy ratio (NER) per 1000 kcal was calculated [36]. This allowed for an objective comparison of the nutrients content of menus with different energy levels. The recommended nutrient-to-energy ratio was calculated on the basis of the age-specific nutritional standards [13].

According to the current nutritional recommendations for children [37], The National Food and Nutrition Institute recommendations for the prevention of diet-related diseases [13,38] and data on nutritional deficiencies of Polish preschoolers [2], the nutrients determining the quality of menus were selected. As key for the quality of nutrition in preschools, the following were selected: energy, total fat, saturated fatty acids, sucrose, dietary fiber, calcium, iron, iodine, potassium, vitamin D and folate. Accordingly, to the Polish recommendation, the full-board preschool menu should provide at least 70% of the daily energy and nutrients requirements of attending children [2], this level was adapted for energy, dietary fiber, vitamins and minerals. Because both a deficiency and an excess of energy can have adverse health consequences, a 10% margin of error from the recommended value was adopted in the case of energy supply. In the case of fat, saturated fatty acids and sucrose, the recommended share of energy from these nutrients was used. Due to the fact that in Polish preschools, the majority are children aged 4–6 years, dietary reference intakes for this age group were adopted. The percentage of preschools implementing these recommendations before and after education was analyzed.

### 2.5. Statistical Analysis

All data were processed statistically using Statistica version 13.1 (Copyright©StatSoft, Inc., 1984–2014, StatSoft Polska Sp. z o.o. manufacturers, Cracow, Poland). The Shapiro–Wilk statistical test for testing the normality of quantitative variables (energy and nutrients content) was used. The amounts of energy and nutrients in menus and nutrients density before and after education were presented as mean and standard deviation. Distribution characteristics of the analyzed nutrients including median and 25th and 75th percentile is presented in the Appendix A. The change (∆ %) in the supply of energy and nutrients and nutrients density before vs. after education was presented as a percentage of the mean initial value. To test the significant differences in the average amounts of energy and nutrients and nutrients density before vs. after the education, a paired student t-test was used for variables with normal distribution or a Wilcoxon signed-rank test for not normally distributed variables. To determine the effect of education on the implementation of the recommendations on the nutrients supply by DCCs, a Pearson’s chi-square test was used. Additionally, contingency coefficient Cramér’s V was used to indicate the strength of association between categorical variables. The differences were considered significant at *p* < 0.05.

## 3. Results

### 3.1. General Characteristic of Preschools

The 231 preschools involved in this study constituted about 21% of directly participating preschools and 3.3% of all government/municipal-sponsored (public) preschools in Poland [4]. The budget for purchasing of food products for full day boarding of one child in preschools ranged from 3.40 to 9.50 PLN with an average of 5.77 ± 1.16 PLN (approximately 1.33 euro). In 87% preschools, a purchasing manager was responsible for menu planning and in 6.5%, a dietitian or kitchen personnel was responsible for menu planning. On average, the preschool provided full board for 127 children (min. 21; max. 411); during the analyzed period, 29,112 children were enrolled in all the examined institutions.

### 3.2. Energy and Nutrients Content of Prechools Menus

Table 1 presents the age-specific dietary reference intake standards, the amounts of energy and macronutrients provided in preschool menus before and after education, and the magnitude of change expressed as a percentage of the average initial value. The quantities of the examined nutrients were diverse, as indicated by the values of the lower and upper quartiles (Appendix A). Assuming that the full-board preschool menu should provide about 70% of the daily demand, in the majority of institutions, children received the correct or higher than recommended amount of energy and macronutrients before education (only the amount of dietary fiber was lower than recommended for 4–6 year-old children in one preschool before education and in three—after). A high share of energy from total fat and saturated fatty acids as well as high protein content were observed (however, the share of energy from protein did not exceed the recommendation). After education, the energy value of the diet, total carbohydrates and total fat as well as the percentage of energy from fat and MUFA decreased significantly. At the same time, an increase in the share of energy from protein was observed.

The overall vitamin composition of examined menus and the magnitude of change expressed as a percentage of the average initial value, as well as the dietary reference standards, are outlined in Table 2. Moreover, in the case of vitamins content, a large variation was observed before and after education, as indicated by 25th and 75th percentile values (Appendix A). Before education, the analyzed menus in most preschools covered at least 70% of the preschoolers’ daily recommended intake for all vitamins except vitamin D (none of the preschools achieved the recommended amount). In eight preschools menus, the supply of vitamin E was too low, in five, the content of folate, and in one—niacin. Education significantly increased the content of vitamin A and beta-carotene on the menu, whereas the amount of vitamin D decreased. Education activities failed to improve the number of preschools that implemented the recommendations for folate and niacin and the number of institutions offered a too low amount of vitamin E doubled.

Table 3 presents the age-specific minerals dietary reference intake standards, the amounts provided in preschool menus before and after education, and the magnitude of change expressed as a percentage of the average initial value. No significant impact of education in the case of minerals supply in preschools menus was observed. The supply of copper, phosphorus, sodium and zinc was at least at the recommended level, both before and after education. The calcium content in the menus was very low: only three preschools menus met the recommendations for this nutrient for 4–6 year-old children before education, and after education, this number increased to seven. In the case of iron, nearly two-thirds (*n* = 150) of preschools menus failed to meet the recommendation for children aged 4–6 years and after education, this number decreased by seven (*n* = 143). A lower than recommended level of potassium (62% menus) and iodine (21% menus) was observed before education. In this case, education increased the percentage of preschools below the recommendation to 29% for potassium and 23% for iodine.

### 3.3. Vitamins and Minerals Density of Prechools Menus

Table 4 presents the nutrient-to-energy ratio for selected vitamins and minerals in relation to the recommended values and the magnitude of change in the average nutrient density of menus before and after education. Before education, the average nutrient density for most vitamins (except vitamin D for both age groups) and minerals (except calcium, iron and potassium for all age groups, and vitamin E and iodine for 1–3 year-old children) was higher than recommended. The beneficial effect of education was observed for most nutrients: vitamins: A, B_1_, B_2_, B_6_, C, folate, and minerals: calcium, copper, iron, magnesium, phosphorus, potassium and zinc. However, even after education, the nutrient-to-energy ratio for shortage nutrients did not reach the recommended values.

### 3.4. Nutrients Important for the Diet Quality

A comparison of the number of preschools serving menus in line with nutritional recommendations important for the overall diet quality before and after education is outlined in Table 5. Prior to education, preschool menus were characterized by a good content of dietary fiber and folate. Over 70% of preschools children received iodine, potassium and energy from fat sources in proper amounts. In over half of the institutions, the recommendation regarding sucrose supply was implemented, and in one third for iron. However, in more than 1/5 of preschools, children received a diet with a too high energy content; in the majority of institutions, the share of energy from saturated fatty acids was too high, and the calcium supply too low. A beneficial effect of education (an increasing of the number of preschools implementing the recommendations) was observed for most of the parameters: energy, fat, SFA, sucrose, calcium and iron. On the other hand, the number of institutions implementing iodine and potassium recommendations decreased significantly after education.

## 4. Discussion

In the case of children attending full-time preschools, the quality of nutrition in these institutions largely determines the correctness of their diet, as they consume up to 75% of the required dietary intake [2,39]. All activities aimed at improving the quality of nutrition in preschools might impact the diet quality of many children. Therefore, it is crucial that the menu offered to children in preschools be properly balanced, especially in terms of nutrients that usually occur in abnormal quantities in the average child diet [39]. The menu offered should not duplicate the typical dietary errors of the population concerned; it should be a good source of scarce nutrients, while the content of those in excess should be limited. However, the implementation of these recommendations in childcare institutions is difficult and not always effective [5,7,10]. Preschools declare various types of difficulties that hinder the implementation of the proper nutrition recommendations in practice [40,41]. These include financial constraints, but also insufficient staff qualifications. Therefore, educational activities increasing personnel knowledge can improve the quality of nutrition in childcare institutions [42]. The Eating Healthy, Growing Healthy program was multicomponent: it included lectures on general topics (e.g., principles of proper nutrition of children) and more detailed issues (e.g., regarding recommendations for sugar, vitamin D, calcium, recommended beverages, etc.), as well as individual analyses of preschool menus with feedback, and provided the possibility of individual educator support for all participating institutions. Therefore, we expected an improvement in the overall quality of menus served in preschools, as well as an increase in the number of institutions implementing nutritional recommendations. In a previously publish paper [30], we have shown the impact of education on the supply of food groups, whereas in the present study, we focused on the energy and nutrients.

The beneficial effects of education included a decrease in the total energy supply and energy from fats. The preschool menu should provide at least 70% of the daily energy demand and all DCCs have implemented this recommendation. However, in the case of energy, not only is shortage unfavorable, but also a too high supply. Excessive energy intake is associated with the risk of overweight and obesity; excessive body mass increases the emergence of comorbidities, including hypertension, type 2 diabetes mellitus, obstructive sleep apnea, non-alcoholic fatty liver disease, and dyslipidemia [43,44]. The excessive consumption of energy in Poland is a bigger problem than its deficiency. Obesity in the Polish pediatric population increased significantly during the last 46 years [45]. A nationally representative study on preschool-aged children showed that obesity rates among Polish 5-year-old children were significantly higher than for their Norwegian peers [46]. In the case of energy, the recommendations apply only to the minimum level (70%), which caused a large range in the energy value of preschool menus. More than 3/4 of preschools served menus with a too high energy content; after education, this number decreased only by 4 percentage points. The hypothetical reason for offering menus with a higher energy value may be the fear that a fussy child may not consume enough energy and feel hungry. However, this approach can promote over-consumption of energy among children with a better appetite. Doubling the portions might increase energy intake by 24% and increasing meal energy density by 42% simultaneously increased energy intake by 40% in preschool children [47]. Our study, the first on such a large number of preschools in Poland, points to the problem and indicates an urgent need to introduce a mandatory range (not only the minimum value) for energy in preschool menus.

Also beneficial is the decrease in fat supply, whereas no change in the share of SFA and a decrease in MUFA is rather a failure of education. Despite the current recommendations of the Polish Institute of Food and Nutrition [13], a typical Polish diet is still rich in fats, especially of animal origin, and consequently, in saturated fatty acids. A study on a representative group of Polish adults indicated that over 80% of men and 70% of women consume excessive amounts of fat and saturated fat [48], which is associated with an increased risk of cardiovascular disease [48,49]. On the other hand, the best food source of MUFA, olive oil, is not typically used in the Polish cuisine. A systematic review and meta-analysis of cohort studies linked higher intakes of olive oil with reduced risk of all-cause mortality, cardiovascular events, and stroke [50]. Therefore, more attention should be paid to promoting this source of fat, especially when reducing saturated fat in the preschool menu. Although no significant reduction in the share of SFA and sucrose were observed, the percentage of preschools implementing dietary recommendations for these macronutrients in their menus increased significantly.

The decrease in the energy content of the menus after education was accompanied by an increase in nutrient density for the most vitamins (except of vitamin C, D, E and niacin). This favourable trend suggests that preschools have implemented the knowledge on products with a higher nutritional value, but it should be emphasized that the baseline supply of these vitamins (except for vitamin D) in the majority of menus was at the correct level. Analyses of young children’s intake in Poland indicated a risk of deficiencies for vitamin D [51,52], vitamin E [51,52] and folate [12], and in the case of these vitamins, education did not cause the expected effect. Our study showed that in the case of folate, nutrient density increased significantly, but the percentage of institutions with the adequate supply did not change. Inadequate folate intake may accordingly be a contributing factor to poor growth [53]. However, an inadequate supply of folate in preschools menus is rather a minor problem, as only a few preschools (5 out of 231) had a problem with achieving the right amount of folate in menus. Vitamin E shortage in preschools menus is also a marginal problem (8 out of 231), unfortunately not solved by education. Reducing the supply of vitamin E may be the result of reducing the supply of fat, but this requires further analyses. However, vitamin E deficiencies in the general population are observed very rarely [54]. The unexpected and most adverse effect of education was related to vitamin D. None of the menus provided adequate amounts of vitamin D before education; moreover, after education, both the amount and nutrient density of vitamin D significantly decreased. The problem of an insufficient intake of vitamin D is common both nationally and internationally and affects both children [55,56] and adults [57]. The actual Polish reference (adequate intake 15 µg/day compared to 5 µg/day in previous Polish standards from 2012 [38]) for vitamin D is very difficult to reach on a daily basis with typical food products. Due to the common deficiencies, supplementation for all population groups throughout the year or only in the period of reduced sunlight in Poland is recommended [58]. However, despite the recommendations, supplementation is not yet common and only 9.1% of adult Poles demonstrated an optimal status of vitamin D [59]. In the case of vitamin D, the effect of education is not only unfavourable but also puzzling, especially since this issue has been given a lot of attention (e.g., a separate topic of lectures and workshops, information about the content of vitamin D in the menu was included in each feedback along with suggestions for improving the situation). Finding the causes of low supply of vitamin D in menus requires further research. As vitamin D has pleiotropic effects [60], the obligatory introduction of fortified products to preschool menus should be considered as a public strategy to increase the dietary intake.

Unfortunately, education did not improve the supply of minerals in the analyzed menus. This was particularly unfavourable for calcium, iron, potassium and iodine, as not all institutions reached the recommendations before education. After education, the nutrient density of all these minerals increased, except for iodine (in this case, no effect was noted). The percentage of preschools implementing the recommendations increased for calcium and iron, while it decreased for potassium and iodine. Although most of the care institutions achieved adequate amounts of potassium in their menus, the adverse effect of education is surprising. Potassium is quite common in food products, and its good sources include legumes, grain products, nuts, dried fruits, selected vegetables and fruits as well as meat and fish [35]. All these products are recommended for proper nutrition of children, and in education, preschools were additionally encouraged to increase the supply of legumes, whole grain product, nuts, vegetables and fruits on the menu. Since other authors also indicated a lower level of potassium in the diet than recommended [55,61,62], more attention should be paid to this nutrient in further educational activities. Similarly, the adverse effect (measured as the percentage of institutions implementing the recommendations) of education was noted for iodine. Even mild iodine deficiency may negatively affect cognitive performance in children [63] and iodine deficiencies are a global health problem. According to the World Health Organization (WHO), 35% of the world’s population has insufficient iodine intake [64]. After twenty years of mandatory iodization of household salt, Poland remains iodine sufficient, but particularly vulnerable populations (school-age children, pregnant and breastfeeding women) are still at risk [65]. Due to the recommendations of dietary salt restriction [37], in our education, we suggested including iodine-rich foods (e.g., fish) on menus. Unfortunately, the supply of fish significantly decreased after education [30], which was reflected in the content of iodine in menus. Since other authors also reported a problem of insufficient iodine intake in preschoolers [61], it seems that the obligatory supply of at least one serving of fish per week should be increased. In the case of iron and calcium, education not only positively increased their amounts per 1000 kcal of diet, but also the percentage of preschools reaching the recommended supply. However, it should be emphasized that in both cases, it is difficult to consider the effect of education as a success since after the training, 38% of menus implemented the recommendations for iron but only 3% for calcium. Due to the recommendations for limiting red meat [37], in our education, we recommended to increase the supply of vegetable origin iron-rich foods (e.g., legumes, nuts, fish). This seems to be a good strategy because, despite a significant drop in meat supply [30], iron density on menus increased. Our research indicates that the biggest educational challenge might be increasing the supply of calcium to recommended values. This is particularly important as an inadequate intake is very often reported in children in Poland [51,55,61,62]. An adequate intake of calcium is crucial for proper bone development [66] and calcium deficiency increases the risk of osteoporosis, which is aggravated by a low intake of vitamin D [67]. The preschools received full information on the adequate supply of calcium, its sources (calcium-rich foods), along with the recommended amounts, as well as a personalized assessment of the menu for its contents. Moreover, the mandatory regulations require the supply of at least two servings of dairy products per day [14]. Despite this, over 90% of preschools did not comply with calcium recommendations. This points to the need to establish not only the number of servings but also their sizes.

### Strengths and Limitations of the Study

Our study provides a unique insight into the effectiveness of nutritional education on the quality of the menus offered to children aged 3–6 years old in preschools in Poland. However, some limitations should be addressed: 1. the survey covered only public (government-sponsored) preschools, so the results cannot be extrapolated to other types of DCCs (i.e., private ones); 2. the study was focused on energy and nutrient supply on an institutional level (menu analyses), not the intake by children; 3. the effect of education was assessed after 3-6 months, so we do not know the long-term effect (e.g., after a year). We performed a univariate statistical analysis of the impact of education. Due to the ethical regulations we did not have access to DCCs employees’ personal data (e.g., professional experience, duration of employment). In addition, employment of a dietitian could be a differentiating factor. As only a very small number of preschools employed a dietitian (only 15 institutions out of 231), we decided to exclude this factor in the statistical analysis. Moreover, the number of staff members varied according to the size of the preschools. As only one person is always responsible for menu planning, we did not include the number of staff in statistical analyses. Additionally, other factors, such as the preference of regional cuisine, availability or preference of local/regional food products, economic factors or equipping with suitable kitchen equipment could also have impacted the results of the effectiveness of education.

A major strength of our study is examining the effect of the multicomponent education program conducted by trained educator on a large sample of preschools located throughout Poland (however, the sample selection was not random). Standardized educational activities were carried out in preschools with a very different initial quality of nutrition, but in each institution, the observed change was related to the baseline value. The effectiveness of education was measured by objective indicators, i.e., changes in the content of energy and nutrients in menus. The content of energy and nutrients was calculated as an average based on 10 consecutive days, and daily inventory reports were used for analysis. The inventory reports show the exact amounts of products used to prepare meals for a specific number of children. The menus, which are often used by other authors, are not fully reliable as they do not always reflect the meals actually served to children [68].

## 5. Conclusions

In our study, the multicomponent education program resulted in some beneficial changes (e.g., improving the nutrient density for calcium and iron), but the scope and magnitude of these changes cannot be considered as fully satisfactory. In our opinion, education alone may not be sufficient to achieve a measurable improvement in the overall diet quality in childcare facilities. Our study highlights the need for uniform mandatory standards of nutrition in preschools, based on the current dietary reference intake standards for population age groups. The US experience showed that within a year after enactment of state and federal policies, 23% of DCCs have fully implemented the law [69]. Therefore, educational activities should support preschools’ personnel in the implementation of mandatory nutrition standards. Mandatory employment of a dietitian in childcare institutions would also be a good step to optimize the quality of nutrition.

## Figures and Tables

**Figure 1 nutrients-11-02438-f001:**
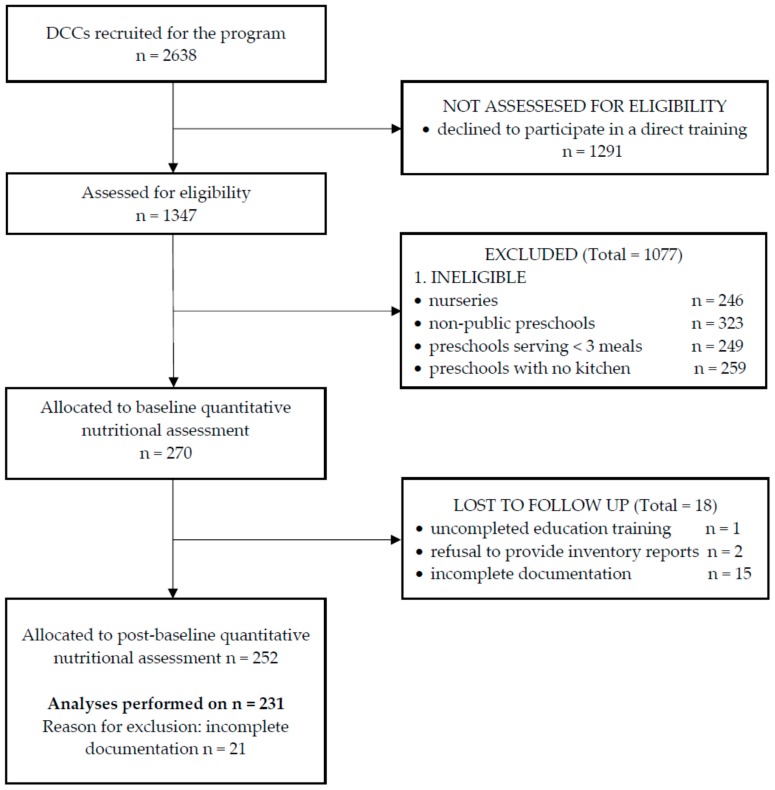
The sampling scheme of the present study.

**Figure 2 nutrients-11-02438-f002:**
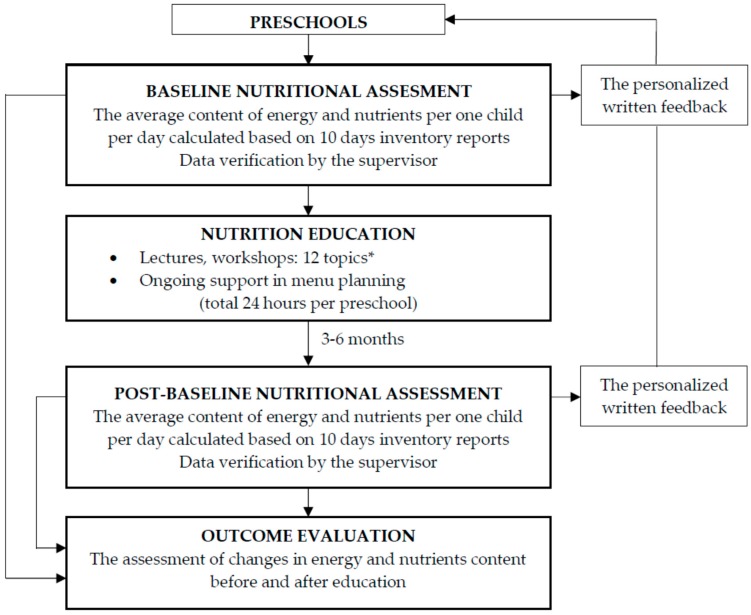
The study design for analyzing the effectiveness of education on the quality of menus in preschools. * The topics of lectures/workshops included 1. General recommendations of balanced nutrition, 2. Water and its role in children’s diets, 3. Sugar in children’s diets, 4. How to deal with food allergies? 5. Fussy eater or food neophobic? 6. Strong bones and teeth—the role of vitamin D and calcium, 7. Salt in children’s diets, 8. How to serve meals attractively, 9. Plump, overweight, obese—how to recognize the problem? 10. Limits of child’s choice, 11. How to involve kids in cooking? 12. Servings size: self-eating, self-deciding.

**Table 1 nutrients-11-02438-t001:** The content of energy and macronutrients (per a child a day) in 231 preschools’ 10-day menus with reference to the age-specific dietary reference intake (DRI) before (baseline) and after (post-baseline) the education program.

Nutrient	DRI for Children Aged 3–6 Years	Daily Supply (per Child/Day)				∆ % ^1^/ *p*-Value *
		Mean ± SD		% of DRI for Children Aged 3–6 Years		
		Baseline	Post-Baseline	Baseline	Post-Baseline	
Energy [kcal]	1000 ^2^/1400 ^3^	1257.1 ± 203.07	1224.3 ± 217.11	126 ^2^/90 ^3^	122 ^2^/87 ^3^	−2.7/0.006
Protein [g]	14 ^2^/21 ^3^	45.8 ± 5.55	45.2 ± 7.61	327 ^2^/218 ^3^	323 ^2^/215 ^3^	−1.4/NS
Protein [% of energy]	10–20	14.6 ± 1.64	14.8 ± 1.30	-	-	1.5/<0.001
Animal protein [g]	NA	28.0 ± 6.68	27.2 ± 5.23	NA	NA	−2.8/NS
Vegetable protein [g]	NA	17.8 ± 3.41	17.9 ± 3.59	NA	NA	1.0/NS
Fat [g]	33–44 ^2^/31–54 ^3^	46.5 ± 9.75	44.7 ± 10.49	141–106 ^2^/150–86 ^3^	135–101 ^2^/144–83 ^3^	−4.0/0.002
Fat [% of energy]	20–35	33.2 ± 3.96	32.7 ± 4.05	-	-	−1.4/0.049 **
SFA [% of energy]	as low as possible	12.9 ± 1.75	12.8 ± 1.70	-	-	−0.2/NS
MUFA [% of energy]	NA	13.1 ± 2.11	12.8 ± 2.14	NA	NA	−2.8/0.006
PUFA [% of energy]	NA	4.8 ± 1.00	4.7 ± 0.94	NA	NA	−1.2/NS
Cholesterol [mg]	<300	170.3 ± 37.02	167.1 ± 37.27	56	56	−1.9/NS
CHO [g]	130	162.6 ± 28.76	159.0 ± 30.36	125	122	−2.3/0.012
CHO [% of energy]	50–70	51.8 ± 3.85	52.0 ± 3.88	104–74	104–74	0.4/NS
Sucrose [% of energy]	NA	9.2 ± 2.89	8.9 ±2.83	NA	NA	−3.4/NS
Lactose [g]	NA	10.6 ± 3.28	10.9 ± 3.32	NA	NA	3.2/NS
Starch [g]	NA	95.8 ± 18.36	94. ± 5 19.98	NA	NA	−1.3/NS
Dietary fiber [g]	10 ^2^/14 ^3^	16.1 ± 3.61	16.4 ± 4.07	161 ^2^/115 ^3^	164 ^2^/117 ^3^	2.3/NS

^1^ the change in the mean supply of energy and nutrients before vs. after education presented as a percentage of the initial value; ^2^ for children aged 3 years; ^3^ for children aged 4–6 years; SFA saturated fatty acids; MUFA monounsaturated fatty acids, PUFA polyunsaturated fatty acids; CHO carbohydrates; NA not available; * significant difference (the Wilcoxon signed-rank test) before vs. after education; ** significant difference (the paired student *t*-test) before vs. after education; NS, not significant.

**Table 2 nutrients-11-02438-t002:** The content of vitamins (per a child a day) in 231 preschools’ 10-day menus with reference to the age-specific dietary reference intake (DRI) before (baseline) and after (post-baseline) the education program.

Nutrient	DRI for Children Aged 3–6 Years	Daily Supply (per Child/Day)				∆ % ^1^/ *p*-Value *
		Mean ± SD		% of DRI for Children Aged 3–6 Years		
		Baseline	Post-Baseline	Baseline	Post-Baseline	
Vitamin A [μg]	400 ^2^/450 ^3^	1070.5 ± 410.37	1121.0 ± 473.16	268 ^2^/238 ^3^	280 ^2^/249 ^3^	4.5/0.043
Retinol [μg]	NA	336.9 ± 260.97	333.6 ± 286.69	NA	NA	−1.0/NS
Beta-carotene [μg]	NA	4398.3 ± 1721.02	4720.4 ± 2012.77	NA	NA	6.8/0.004
Vitamin B_1_ [mg]	0.5 ^2^/0.6 ^3^	0.9 ± 0.17	0.9 ± 0.16	180 ^2^/150 ^3^	180 ^2^/150 ^3^	−0.3/NS
Vitamin B_2_ [mg]	0.5 ^2^/0.6 ^3^	1.1 ± 0.21	1.1 ± 0.21	220 ^2^/183 ^3^	220 ^2^/183 ^3^	0.8/NS
Vitamin B_6_ [mg]	0.5 ^2^/0.6 ^3^	1.6 ± 0.30	1.6 ± 0.30	320 ^2^/267 ^3^	320 ^2^/267 ^3^	−0.6/NS
Vitamin B_12_ [μg]	0.9 ^2^/1.2 ^3^	2.9 ± 1.16	2.9 ± 1.26	322 ^2^/167 ^3^	322 ^2^/167 ^3^	−1.8/NS
Vitamin C [mg]	40 ^2^/50 ^3^	101.9 ± 29.81	105.9 ± 38.58	255 ^2^/204 ^3^	265 ^2^/212 ^3^	3.8/NS
Vitamin D [μg]	15	1.8 ± 0.63	1.7 ± 0.59	12	11	−6.4/0.032
Vitamin E [mg]	6	6.6 ± 1.78	6.4 ± 1.60	110	107	−3.0/NS
Folate [μg]	150 ^2^/200 ^3^	221.1 ± 47.71	223.8 ± 49.46	147 ^2^/111 ^3^	149 ^2^/112 ^3^	1.2/NS
Niacin [mg]	6 ^2^/8 ^3^	10.3 ± 2.01	10.2 ± 1.94	172 ^2^/129 ^3^	170 ^2^/128 ^3^	−1.6/NS

^1^ The change in the mean supply of vitamins before vs. after education presented as a percentage of the initial value; ^2^ for children aged 3 years; ^3^ for children aged 4–6 years; NA not available; * significant difference (the Wilcoxon signed-rank test) before vs. after education; NS, not significant.

**Table 3 nutrients-11-02438-t003:** The content of minerals (per a child a day) in 231 preschools’ 10-day menus with reference to the age-specific dietary reference intake (DRI) before (baseline) and after (post-baseline) the education program.

Nutrient	DRI for Children Aged 3–6 Years	Daily Supply (per Child/Day)				∆ % ^1^/ *p*-Value
		Mean ± SD		% of DRI for Children Aged 3–6 Years		
		Baseline	Post-Baseline	Baseline	Post-Baseline	
Calcium [mg]	700 ^2^/1000 ^3^	454.3 ± 108.56	463.6 ± 110.41	65 ^2^/45 ^3^	66 ^2^/46 ^3^	2.0/NS
Copper [mg]	0.3 ^2^/0.4 ^3^	0.9 ± 0.27	0.9 ± 0.17	300 ^2^/225 ^3^	300 ^2^/225 ^3^	−0.3/NS
Iodine [μg]	90	108.0 ± 64.48	104.1 ± 56.47	120	116	−3.8/NS
Iron [mg]	7 ^2^/10 ^3^	6.7 ± 1.42	6.7 ± 1.33	96 ^2^/67 ^3^	96 ^2^/67 ^3^	0.3/NS
Magnesium [mg]	80 ^2^/130 ^3^	206.2 43.08	207.0 ± 38.55	258 ^2^/159 ^3^	259 ^2^/159 ^3^	0.4/NS
Phosphorus [mg]	460 ^2^/500 ^3^	802.4 ± 139.35	809.0 ± 138.66	174 ^2^/160 ^3^	176 ^2^/162 ^3^	0.8/NS
Potassium [mg]	2400 ^2^/3100 ^3^	2469.0 ± 472.67	2452.8 ± 480.32	103 ^2^/80 ^3^	102 ^2^/79 ^3^	−0.7/NS
Sodium [mg]	750 ^2^/1000 ^3^	2288.0 ± 1164.02	2192.6 ± 1115.27	305 ^2^/229 ^3^	292 ^2^/219 ^3^	−4.3/NS
Zinc [mg]	3 ^2^/5 ^3^	5.9 ± 1.10	6.0 ± 1.05	197 ^2^/118 ^3^	200 ^2^/120 ^3^	0.8/NS

^1^ the change in the mean supply of minerals before vs. after education presented as a percentage of the initial value; ^2^ for children aged 3 years; ^3^ for children aged 4–6 years; NS, not significant.

**Table 4 nutrients-11-02438-t004:** The nutrient-to-energy ratio (NER) in 231 preschools’ 10-day menus with reference to the recommended age-specific nutrient-to-energy ratio before (baseline) and after (post-baseline) the education program.

Nutrient	Recommended NER ^1^		Preschools Menus NER ^2^		∆ % ^3^/ *p*-Value *
	Aged 1–3 Years	Aged 4–6 Years	Baseline	Post-Baseline	
			Mean ± SD	Mean ± SD	
Vitamin A [μg]	400	321	853 ± 303.2	927 ± 427.1	8.0/<0.001
Vitamin B_1_ [mg]	0.5	0.4	0.7 ± 0.09	0.7 ± 0.08	2.5/0.002
Vitamin B_2_ [mg]	0.5	0.4	0.9 ± 0.12	0.9 ± 0.13	3.7/< 0.001
Vitamin B_6_ [mg]	0.5	0.4	1.2 ± 0.17	1.3 ± 0.17	2.1/0.023
Vitamin B_12_ [μg]	0.9	0.9	2.3 ± 0.88	2.4 ± 1.11	1.6/NS
Vitamin C [mg]	40	36	81 ± 19.8	87 ± 27.3	6.5/0.002
Vitamin D [μg]	15	11	1.4 ± 0.49	1.3 ± 0.45	−3.8/NS
Vitamin E [mg]	6	4.3	5.2 ± 0.99	5.2 ± 0.89	-
Folate [μg]	150	143	176 ± 26.6	184 ± 31.8	4.2/<0.001
Niacin [mg]	6	5.7	8.2 ± 1.10	8.3 ± 1.08	1.2/NS
Calcium [mg]	700	714	363 ± 75.6	382 ± 79.6	4.8/<0.001 **
Copper [mg]	0.3	0.3	0.7 ± 0.17	0.7 ± 0.10	3.0/<0.001
Iodine [μg]	90	64	86 ± 47.6	85 ± 41.3	−1.6/NS
Iron [mg]	7	7.1	5.3 ± 0.73	5.5 ± 0.72	3.2/<0.001
Magnesium [mg]	80	93	164 ± 24.2	170 ± 20.7	3.2/<0.001
Phosphorus [mg]	460	357	641 ± 70.3	665 ± 75.4	3.7/<0.001
Potassium [mg]	2400	2214	1820 ± 256.2	2014 ± 271.1	2.2/0.018
Sodium [mg]	750	714	1971 ± 851.0	1778 ± 783.7	−2.4/NS
Zinc [mg]	3	3.6	4.7 ± 0.51	4.9 ± 0.52	3.7/<0.001

^1^ The recommended amount of nutrient per 1000 kcal; ^2^ the amount of nutrient per 1000 kcal of served menus; ^3^ the change in mean nutrient-to-energy ratio before vs. after education presented as a percentage of the initial value; * significant difference (the paired Wilcoxon signed-rank test) before vs. after education; ** significant difference (the student *t*-test) before vs. after education; NS, not significant.

**Table 5 nutrients-11-02438-t005:** The adequate supply of nutrients important for the overall diet quality in 231 preschools’ 10-day menus before (baseline) and after (post-baseline) the education program.

Nutrient	Recommendations for Preschool Menu ^1^	Preschools Meeting the Recommendations		*p*-Value */Cramér’s V
		number/%		
		Baseline	Post-Baseline	
Energy [kcal]	880–1080	50/21.6	60/26.0	<0.001/0.32
Total fat [% of Energy]	20–35%	163/70.6	171/74.0	<0.001/0.31
SFA [% of Energy]	<10% ^2^	6/2.6	11/4.8	<0.001/0.52
Sucrose [% of Energy]	<10% ^3^	128/55.4	152/65.8	<0.001/0.45
Dietary Fiber [g]	9.8	230/99.6	228/98.7	NS
Calcium [mg]	700	3/1.3	7/3.0	0.002/0.20
Iron [mg]	7	81/35.1	88/38.1	<0.001/0.50
Iodine [μg]	63	182/78.8	178/77.0	<0.001/0.34
Potassium [mg]	2170	169/73.2	163/70.0	<0.001/0.41
Vitamin D [μg]	10.5	0/0	0/0	-
Folate [μg]	140	226/97.8	226/97.8	-

^1^ 70% of the dietary reference intake for 4–6 year-old children; ^2^ based on the recommendation of The National Food and Nutrition Institute, 2012; ^3^ based on the recommendation of The Polish Expert Group, 2012; * significant differences before vs. after education, chi ^2^ Pearson’s chi-squared test, *p* < 0.05; SFA, saturated fatty acids; NS, not significant.

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
