# Peer review of "Eating Healthy, Growing Healthy: Outcome Evaluation of the Nutrition Education Program Optimizing the Nutritional Value of Preschool Menus, Poland"

_nutrients, 2019, doi:10.3390/nu11102438_

Round 1

Reviewer 1 Report

This paper is a presentation of a large-scale intervention, however, the manuscript as presented does not do the work justice. The authors present an introduction that is dominated by assumptions and conjecture and fail to lead the reader to a clear understanding for the rationale and the interpretation of results. Results are presented poorly. The language is, in part, judgemental and needs to be reworded (i.e. "low quality evidence" in line 484).

Introduction:

provide a rationale, for instance by answering basic questions

what is the average nutrient and energy intake of preschool age children in poland? is there cause to believe that diet-related chronic diseases are a public health issue? why is an examination of dietary intake patterns deemed necessary? are other issues, such as SES or family food environment a concern? in how far do menu analysis reflect children's intake?

please clarify the terminology, i.e. "full board centers"

Results:

consider rather than presenting raw data, convert nutrient intakes to ratios indicating the actual/recommended values to add clarity

Table 1-4, remove median and percentile ranges. since only menu analysis (not actual food intake) are presented the mean and SD and their comparison is sufficient.

to cut the amount of statistical tests, present either nutrients in grams or in %total energy

present delta of change and p-values, remove arrows

select important nutrients and present in one table (rather than presenting vitamins separate from macronutrients etc.)

Table 5, the only really valuable table because it allows easy comparison between the pre- and post values, however, remove the zeros of the p-values

the authors should consider reducing the number of statistical tests, alternatively a correction of the p-value is necessary

Discussion and conclusion

the discussion is very wordy and needs to be significantly shorter.

one main concern is that the authors discuss at great length the changes in the menus, however, as stated in the introduction, the overall goal is to improve child nutrition. No food acceptance data was presented. Thus, the conclusion must be reworded to clearly state that although the intervention was successful in changing the foods listed on the menus, the extend to which those changes may actually contribute to the improvement of dietary intake is not known. 

the authors should remove concepts, such as "children's behavior change" etc. since the data collected did not include any individual- or group level behavioral or dietary data. the outcome variables of this study were based on the effectiveness of teaching how to improve diet quality of menus, no further conclusions can be drawn.

Author Response

Response to Reviewer 1 Comments

We appreciate the time spent and all comments made to improve the manuscript.

We've provided answers to all comments below. The changes made to the manuscript in response to comments from Reviewer 1 are highlighted in green.

Comments and Suggestions for Authors

This paper is a presentation of a large-scale intervention, however, the manuscript as presented does not do the work justice. The authors present an introduction that is dominated by assumptions and conjecture and fail to lead the reader to a clear understanding for the rationale and the interpretation of results. Results are presented poorly. The language is, in part, judgemental and needs to be reworded (i.e. "low quality evidence" in line 484).

Introduction:

R comment: provide a rationale, for instance by answering basic questions

what is the average nutrient and energy intake of preschool age children in poland? is there cause to believe that diet-related chronic diseases are a public health issue? why is an examination of dietary intake patterns deemed necessary? are other issues, such as SES or family food environment a concern? in how far do menu analysis reflect children's intake?

Authors: We cannot fully agree with the comment. In our opinion, the Introduction is adequate to the presented content, i.e. it presents the issues related to the implementation of nutrition in childcare facilities in Poland and internationally. The introduction clearly indicated existing irregularities in the implementation of nutrition in DCCs, which suggests the need to improve this situation, e.g. through education of staff. We believe that information on the analysis of nutrition patterns, SES or family food environment are unrelated to the topic and scope of work as we do not focus on children intake (neither in preschools nor at home environmental). SES or family food environment are not related to the implementation of nutrition in DCCs.

In contrast, data on nutrients intake as well as information on the risk of diet-related chronic diseases (e.g. energy consumption and risk of obesity) are presented in the discussion. Presenting this data in a discussion allows linking existing irregularities and effects of education program with the overall child's nutritional status.

However, following the comment, we have added information on how far menu analysis reflect children's intake.

R comment: please clarify the terminology, i.e. "full board centers"

Authors: There is no a strict definition for “full board centers”, but usually full board preschools provide two main meals and 1-2 snack a day. It has been explained on P4 but following the suggestion this information has been added in Introduction.

Results:

R comment: consider rather than presenting raw data, convert nutrient intakes to ratios indicating the actual/recommended values to add clarity

Authors: Raw data (values) are necessary for international comparisons, as intake recommendations can vary considerably. But following the comment, we have added data on the implementation of the recommendations to the tables.

R comment: Table 1-4, remove median and percentile ranges. since only menu analysis (not actual food intake) are presented the mean and SD and their comparison is sufficient.

Authors: We believe that presenting the value of quartiles better characterizes the large variation in the content of nutrients in preschool menus. Moreover, in the absence of normal distribution (as in most of analyzed nutrients) it is rather recommended to use the median as more accurate than mean value (Motulsky HJ. Common misconceptions about data analysis and statistics. Br J Pharmacol. 2015; 172(8):2126-32). However, following the comment we propose to move this data to Supplementary Materials.

R comment: to cut the amount of statistical tests, present either nutrients in grams or in %total energy

Authors: We cannot follow this comment because for these few nutrients (protein, fat, sucrose) Polish nutritional standards are expressed as a percentage of energy. For example, at higher caloric content, the saturated fatty acid content may be higher - therefore these components should not be presented in grams.

R comment: present delta of change and p-values, remove arrows

Authors: This has been changed.

R comment: select important nutrients and present in one table (rather than presenting vitamins separate from macronutrients etc.)

Authors: Typical division used commonly in the literature when discussing nutrients supply/intake include macronutrients, vitamins and minerals separately. The nutrients we discussed are important for the quality of the diet; there are nutritional standards for them. We do not discuss nutrients, e.g. molybdenum or flavonoids, as there are no nutritional standards for them. That is why we prefer to maintain this division: it facilitates understanding and can also be a source of date for comparisons with other countries.

R comment: Table 5, the only really valuable table because it allows easy comparison between the pre- and post values, however, remove the zeros of the p-values

Authors: This has been changed.

R comment: the authors should consider reducing the number of statistical tests, alternatively a correction of the p-value is necessary

Authors: These statistical tests are necessary to confirm (or not) the significance of the change but following the suggestion we made a correction of p-value.

Discussion and conclusion

R comment: the discussion is very wordy and needs to be significantly shorter.

Authors: Unfortunately, this remark is a very general and does not indicate fragments which according to the reviewer are unnecessary (at the same time the second reviewer did not request to shorten the discussion, and even suggested its extension with new elements, e.g. nutritional deficiencies). However, following the suggestion, we made some shortcuts in the manuscript text (marked in the text).

R comment: one main concern is that the authors discuss at great length the changes in the menus, however, as stated in the introduction, the overall goal is to improve child nutrition. No food acceptance data was presented. Thus, the conclusion must be reworded to clearly state that although the intervention was successful in changing the foods listed on the menus, the extend to which those changes may actually contribute to the improvement of dietary intake is not known.

Authors: We cannot fully agree with the above comment. The purpose of the present study was formulated adequately to the scope of the study and reads (Page 2): “Therefore, the purpose of this study was to evaluate the effectiveness of the educational program for improving the nutritional value of preschools menus in Poland measured by the change in nutrients content before and after education.”; Abstract: " This study evaluated the effectiveness of the multicomponent educational program for improving the nutritional value of preschools menus in Poland measured by the change in nutrients content before (baseline) and 3-6 month after education (post-baseline)."

Improving the quality of preschool menus was aimed to improve the quality of children's nutrition. Studies show that children consume 50 to 100% of what is offered to them in day care. At the same time, nutrition in preschools covers up to 75% of their daily needs. So, if the menu in the care centers is unbalanced, it greatly affects the correctness of their nutrition.

R comment: the authors should remove concepts, such as "children's behavior change" etc. since the data collected did not include any individual- or group level behavioral or dietary data. the outcome variables of this study were based on the effectiveness of teaching how to improve diet quality of menus, no further conclusions can be drawn.

Authors: We cannot fully agree with the above comment. Our conclusions do not relate to children's nutritional behavior, but only relate to the quality of nutrition in care facilities (section Conclusions, pages 18-19: "In our study, the multicomponent education program resulted in some beneficial changes, but the scope and magnitude of these changes cannot be considered as fully satisfactory. In our opinion, education alone may not be sufficient to achieve a measurable improvement in the overall diet quality in childcare facilities...."). However, in this part we referred to other interventions aimed at improving the quality of children's nutrition, which also includes eating behaviors. However, following the comment we have made changes (marked in green).

Reviewer 2 Report

Thank you for giving me the opportunity to review the article. The authors conducted a study on the outcome evaluation of the nutrition education program optimizing the nutritional value of preschools menus in Poland. I think that the topic is important for the people in the country, but the implication for the readers in other countries should be presented to publish in the international journal (such as Nutrients). The comments for the manuscript were listed below.

Comments:

Introduction:

P1-2: The authors should add the importance of this study for the readers in foreign countries with appropriate references.

Materials and Methods:

P2: The authors should provide the materials for advertisement of this study. P2-3: The details of the program (both two programs) should add as the supplementary material. It because that the potential readers cannot understand the contents of the educational program through the current manuscript. P4 (Figure 1): The flow diagram should be corrected according to the standards for clinical trials. You can check the format in the CONSORT and/or major medical journals (e.g. NEJM and JAMA). When correcting the figure, the authors should add a pair of the reason and the number of centers excluded from the study. P4: Why did the authors not assess the knowledge level of the staff members in the DCCs. It may be an important limitation of this study. P4: Why did the authors assess the effects of the program after 3 to 6 months? This should be justified with relevant references. P6: The authors only performed univariate analysis, but they should consider factors which associate with the nutritional management (e.g. number and experiences (specialty) of staff members).

Discussion:

P13-16: Other factors which can affect the changes of the menus should be discussed in detail. P13-16: The authors did not show any educational contents in the manuscript; therefore, it was difficult to implicate the current results of the nutritional analysis. P13-16: Not only the meeting recommendation, the magnitude of the insufficiency of the menus should be evaluated, and should discuss about the situations in Poland. P13-16: Seasonal differences of the menus should be discussed.

Author Response

Response to Reviewer 2 Comments

We appreciate the time spent and all comments made to improve the manuscript. Particularly helpful was the comment about the diagram (which is now much more readable in our opinion).

We've provided answers to all comments below. The changes made to the manuscript in response to comments from Reviewer 2 are highlighted in blue.

COMMENTS:

Introduction:

P1-2: The authors should add the importance of this study for the readers in foreign countries with appropriate references.

Authors: It was done as suggested. Studies of child nutrition, not only in Poland but also internationally, point to numerous irregularities (some of them are common, e.g. low vegetables intake, low calcium intake). Therefore, a lot of attention is paid to nutrition education. When planning such education activities, information not only about their scope but, above all, the effectiveness of actions taken can be very useful. Such data is presented in our article.

Materials and Methods:

P2: The authors should provide the materials for advertisement of this study.

Authors: Information about the program was sent via e-mail to institutions and placed in available media dedicated to institutions (e.g. educational newsletters). All these materials are only in Polish. The information included the purpose and scope of the project and a link to the website through which the institutions registered themselves to the program (http://zdrowojemy.info/placowka/zapisz-placowke-do-programu). At the moment, the link enabling registration in the program is no longer active, because the program is completed. However, the website with the program description is active and this information has been included in the manuscript text.

Now it reads: The program was nationwide; participation was voluntary and free of charge for participating institutions. The contact details for DCCs were obtained from the local relevant municipal institutions. All registred DCCs received an e-mail invitation to participate, in addition information about the program was advertised in media channels addressed to care and educational centres (press, internet portals, etc.). To participate in the program, the institutions had to register on a dedicated website (http://zdrowojemy.info/placowka/zapisz-placowke-do-programu, in Polish) until the end of the recruitment period. 

P2-3: The details of the program (both two programs) should add as the supplementary material. It because that the potential readers cannot understand the contents of the educational program through the current manuscript.

Authors: A diagram with the overall scheme of the program Healthy Eating, Healthy Growing (Figure S1) is available in supplementary materials. Information on this is included in the Materials and Methods in the section General Information and under the text of the manuscript (Supplementary Materials).

P4 (Figure 1): The flow diagram should be corrected according to the standards for clinical trials. You can check the format in the CONSORT and/or major medical journals (e.g. NEJM and JAMA). When correcting the figure, the authors should add a pair of the reason and the number of centers excluded from the study.

Authors: This was done as suggested. Figure 1 has been modified and information has been added about the number of excluded preschools at each stage along with the reasons for exclusion. Our study is not a RCT but we tried to use the guidelines from the JAMA CONSORT. In addition, we also modified Figure 2, in which we added information on the topics of educational workshops.

P4: Why did the authors not assess the knowledge level of the staff members in the DCCs. It may be an important limitation of this study.

Authors: The scope of the program did not include assessing the knowledge level of employees of the institutions participating in the program. In Poland, there are numerous publications assessing the level of nutritional knowledge of care staff, both preschools and nurseries (e.g. Stankiewicz, Bogdańska: Assessment of the extent of nutritional awareness among kindergarten employees in terms of proper nutrition of preschool children. Probl Hig Epidemiol 2013, 94(3): 479-483; Chalcarz et al. Nutritional knowledge of kindergarten employees. Nowa Medycyna 7/1999, 62-67).

Additionally, before the beginning of the program, an analysis of the level of nutritional knowledge was carried out among a representative group (n = 892) of the caregivers for children aged 3-6 which. Obtained results were the basis for developing the model of nutritional education used in the EHGH project (results of this study were published as a monograph, in Polish; available on-line:

148.81.185.134/exlibris/aleph/a22_1/apache_media/2U8XMSYHMNDJU4S83T33QGE4XFUL3F.pdf).

The program partner (Jan Amos Comenius Children's Development Foundation) has many years of experience in this field and has been cooperating with child care institutions for many years. Both the literature data and the Foundation's experience indicate an insufficient level of nutritional knowledge and the need for staff education, which contributed to the design idea of this educational project.

From a practical reason, with the assumed number of institutions (13 214 nurseries and preschools employees were trained within the EHGH) verification of the knowledge of each participant would not be possible.

Moreover, it seems to us that the level of initial knowledge of the staff could affect the correctness of the implementation of nutrition at baseline (before education), and not the effectiveness of education itself as educational materials were the same for all facilities and the availability of an educator (on-going support) allowed staff to ask additional questions in case of any doubts.

P4: Why did the authors assess the effects of the program after 3 to 6 months? This should be justified with relevant references.

Authors: There are no standards that specify the time required to assess the effectiveness of educational activities. The assessment can be carried out during educational activities (especially if the intervention itself is longer), immediately after completing education or after a longer period, usually 3, 6 or 12 months (ref: DeCosta, et al: Changing children's eating behaviour - A review of experimental research. Appetite 113 (2017) 327e357). In our case, we have adopted a period of 3 to 6 months. On the one hand, this allowed changes to legal contracts with food suppliers, and on the other, it did not allow the implementation of another educational program before the post-baseline menu analysis. Thanks to this strategy, we were able to assess the impact of our educational activities. We add this information to the manuscript and it reads: “Such a period is used in this type of interventions [23]; it allowed institutions to change their food suppliers if necessary, while not allowing the implementation of another educational program.”

P6: The authors only performed univariate analysis, but they should consider factors which associate with the nutritional management (e.g. number and experiences (specialty) of staff members).

Authors: Our study group was homogeneous: only public facilities providing full board and preparing meals from scratch, which is why we conducted univariate analyzes. We did not have the consent of the bioethics commission to collect personnel data (e.g. period of employment or other professional experience). We have only obtained information about employing (or not) a dietitian for menu planning. Only in 15 institutions out of 231 a person with such specialist education was employed, which significantly reduced the validity of the analysis. We agree that having additional information about staff could enable multi-factor analysis.

However, one person is always responsible for menu planning, while the number of people employed in the kitchen may vary (depending on the size of the DCC).

Discussion:

P13-16: Other factors which can affect the changes of the menus should be discussed in detail.

Authors: We have made every effort (homogeneity of the group, identical educational content presented by specially trained educators, specific period of evaluation), so that the observed change was only the effect of our education. Of course, as in the case of this type of research, we cannot say with absolute certainty. However, the data we have does not allow us to discuss the detailed impact of other potential factors. An example would be the impact of regional cuisine (local food available). Following the suggestion, we added these factors to the study limitation (marked in blue in manuscript).

P13-16: The authors did not show any educational contents in the manuscript; therefore, it was difficult to implicate the current results of the nutritional analysis.

Authors: This was done as suggested. The topic of the education activities are now available in the manuscript (under the Figure 2, page 5)

P13-16: Not only the meeting recommendation, the magnitude of the insufficiency of the menus should be evaluated, and should discuss about the situations in Poland.

Authors: It was done as suggested (changes in the manuscript are marked in blue).

P13-16: Seasonal differences of the menus should be discussed.

Authors: Thank you for the suggestion. The available data on the influence of season are very limited in Poland. Moreover, the economic situation (availability of food products) in Poland changed rapidly within last years (from the absolute unavailability of food products to their widespread presence in local markets). We have expected seasonal variations in nutrient content and conducted basic tests for the planned article. However, we did not observe such significance in the content of nutrients. The differences were rather in relation to selected products, i.e. vegetables and fruits (e.g. strawberries were served more often in the summer, and oranges in the winter, both products are a very good source of vitamin C), not nutrients - that is why we gave up preparing manuscript. However, one study of preschools menus from 2014 showed the influence of season in vitamin C content. On the other hand, similar study on hospitals menus did not show any differences in vitamin C and other nutrients content affected by season. Both these studies were conducted in one preschool / hospital and only 10 days menus from 4 season were analyzed (40 days in total). So the results are rather weak for generalization. Due to the reviewers comment (appearing for the second time), we will definitely come back to planned manuscript, showing the lack of seasonal differences in a large group of kindergartens.

Round 2

Reviewer 2 Report

Thank you for giving me the opportunity to review the revised version of this manuscript. Most of the parts pointed out in the first review process were revised appropriately, and the quality of the manuscript was improved. However, several minor revisions should be needed before acceptance. I listed the detailed comments (additional comment: AC) below.

COMMENTS:

Introduction:

P1-2: The authors should add the importance of this study for the readers in foreign countries with appropriate references.

Authors: It was done as suggested. Studies of child nutrition, not only in Poland but also internationally, point to numerous irregularities (some of them are common, e.g. low vegetables intake, low calcium intake). Therefore, a lot of attention is paid to nutrition education. When planning such education activities, information not only about their scope but, above all, the effectiveness of actions taken can be very useful. Such data is presented in our article.

AC: I thought that the point was revised appropriately.

Materials and Methods:

P2: The authors should provide the materials for advertisement of this study.

Authors: Information about the program was sent via e-mail to institutions and placed in available media dedicated to institutions (e.g. educational newsletters). All these materials are only in Polish. The information included the purpose and scope of the project and a link to the website through which the institutions registered themselves to the program (http://zdrowojemy.info/placowka/zapisz-placowke-do-programu). At the moment, the link enabling registration in the program is no longer active, because the program is completed. However, the website with the program description is active and this information has been included in the manuscript text.

Now it reads: The program was nationwide; participation was voluntary and free of charge for participating institutions. The contact details for DCCs were obtained from the local relevant municipal institutions. All registred DCCs received an e-mail invitation to participate, in addition information about the program was advertised in media channels addressed to care and educational centres (press, internet portals, etc.). To participate in the program, the institutions had to register on a dedicated website (http://zdrowojemy.info/placowka/zapisz-placowke-do-programu, in Polish) until the end of the recruitment period.

AC: I thought that the contents was revised appropriately. However, typos should be corrected (e.g. “registred”, “centres”).

P2-3: The details of the program (both two programs) should add as the supplementary material. It because that the potential readers cannot understand the contents of the educational program through the current manuscript.

Authors: A diagram with the overall scheme of the program Healthy Eating, Healthy Growing (Figure S1) is available in supplementary materials. Information on this is included in the Materials and Methods in the section General Information and under the text of the manuscript (Supplementary Materials).

AC: I thought that the contents was revised appropriately. However, a typo should be corrected (e.g. “OUTCOM”).

P4 (Figure 1): The flow diagram should be corrected according to the standards for clinical trials. You can check the format in the CONSORT and/or major medical journals (e.g. NEJM and JAMA). When correcting the figure, the authors should add a pair of the reason and the number of centers excluded from the study.

Authors: This was done as suggested. Figure 1 has been modified and information has been added about the number of excluded preschools at each stage along with the reasons for exclusion. Our study is not a RCT but we tried to use the guidelines from the JAMA CONSORT. In addition, we also modified Figure 2, in which we added information on the topics of educational workshops.

AC: I thought that the point was revised appropriately.

P4: Why did the authors not assess the knowledge level of the staff members in the DCCs. It may be an important limitation of this study.

Authors: The scope of the program did not include assessing the knowledge level of employees of the institutions participating in the program. In Poland, there are numerous publications assessing the level of nutritional knowledge of care staff, both preschools and nurseries (e.g. Stankiewicz, Bogdańska: Assessment of the extent of nutritional awareness among kindergarten employees in terms of proper nutrition of preschool children. Probl Hig Epidemiol 2013, 94(3): 479-483; Chalcarz et al. Nutritional knowledge of kindergarten employees. Nowa Medycyna 7/1999, 62-67).

Additionally, before the beginning of the program, an analysis of the level of nutritional knowledge was carried out among a representative group (n = 892) of the caregivers for children aged 3-6 which. Obtained results were the basis for developing the model of nutritional education used in the EHGH project (results of this study were published as a monograph, in Polish; available on-line: 

148.81.185.134/exlibris/aleph/a22_1/apache_media/2U8XMSYHMNDJU4S83T33QGE4XFUL3F.pdf).

The program partner (Jan Amos Comenius Children's Development Foundation) has many years of experience in this field and has been cooperating with child care institutions for many years. Both the literature data and the Foundation's experience indicate an insufficient level of nutritional knowledge and the need for staff education, which contributed to the design idea of this educational project.

From a practical reason, with the assumed number of institutions (13214 nurseries and preschools employees were trained within the EHGH) verification of the knowledge of each participant would not be possible.

Moreover, it seems to us that the level of initial knowledge of the staff could affect the correctness of the implementation of nutrition at baseline (before education), and not the effectiveness of education itself as educational materials were the same for all facilities and the availability of an educator (on-going support) allowed staff to ask additional questions in case of any doubts.

AC: I understood the situation, and the authors should write about the insufficient level of nutritional knowledge and the need for staff education in the Introduction and/or the Discussion section(s).

P4: Why did the authors assess the effects of the program after 3 to 6 months? This should be justified with relevant references.

Authors: There are no standards that specify the time required to assess the effectiveness of educational activities. The assessment can be carried out during educational activities (especially if the intervention itself is longer), immediately after completing education or after a longer period, usually 3, 6 or 12 months (ref: DeCosta, et al: Changing children's eating behaviour - A review of experimental research. Appetite 113 (2017) 327e357). In our case, we have adopted a period of 3 to 6 months. On the one hand, this allowed changes to legal contracts with food suppliers, and on the other, it did not allow the implementation of another educational program before the post-baseline menu analysis. Thanks to this strategy, we were able to assess the impact of our educational activities. We add this information to the manuscript and it reads: “Such a period is used in this type of interventions [23]; it allowed institutions to change their food suppliers if necessary, while not allowing the implementation of another educational program.”

AC: I understood the situation to decide the time of assessments.

P6: The authors only performed univariate analysis, but they should consider factors which associate with the nutritional management (e.g. number and experiences (specialty) of staff members).

Authors: Our study group was homogeneous: only public facilities providing full board and preparing meals from scratch, which is why we conducted univariate analyzes. We did not have the consent of the bioethics commission to collect personnel data (e.g. period of employment or other professional experience). We have only obtained information about employing (or not) a dietitian for menu planning. Only in 15 institutions out of 231 a person with such specialist education was employed, which significantly reduced the validity of the analysis. We agree that having additional information about staff could enable multi-factor analysis.

However, one person is always responsible for menu planning, while the number of people employed in the kitchen may vary (depending on the size of the DCC).

AC: I understood the reason why did the authors only conducted univariate analysis. However, these contexts should be included in the manuscript for the readers who does not know the situation in Poland.

Discussion:

P13-16: Other factors which can affect the changes of the menus should be discussed in detail.

Authors: We have made every effort (homogeneity of the group, identical educational content presented by specially trained educators, specific period of evaluation), so that the observed change was only the effect of our education. Of course, as in the case of this type of research, we cannot say with absolute certainty. However, the data we have does not allow us to discuss the detailed impact of other potential factors. An example would be the impact of regional cuisine (local food available). Following the suggestion, we added these factors to the study limitation (marked in blue in manuscript).

AC: I thought that the point was revised appropriately.

P13-16: The authors did not show any educational contents in the manuscript; therefore, it was difficult to implicate the current results of the nutritional analysis.

Authors: This was done as suggested. The topic of the education activities are now available in the manuscript (under the Figure 2, page 5)

AC: I thought that the point was revised appropriately.

P13-16: Not only the meeting recommendation, the magnitude of the insufficiency of the menus should be evaluated, and should discuss about the situations in Poland.

Authors: It was done as suggested (changes in the manuscript are marked in blue).

AC: I thought that the point was revised appropriately.

P13-16: Seasonal differences of the menus should be discussed.

Authors: Thank you for the suggestion. The available data on the influence of season are very limited in Poland. Moreover, the economic situation (availability of food products) in Poland changed rapidly within last years (from the absolute unavailability of food products to their widespread presence in local markets). We have expected seasonal variations in nutrient content and conducted basic tests for the planned article. However, we did not observe such significance in the content of nutrients. The differences were rather in relation to selected products, i.e. vegetables and fruits (e.g. strawberries were served more often in the summer, and oranges in the winter, both products are a very good source of vitamin C), not nutrients - that is why we gave up preparing manuscript. However, one study of preschools menus from 2014 showed the influence of season in vitamin C content. On the other hand, similar study on hospitals menus did not show any differences in vitamin C and other nutrients content affected by season. Both these studies were conducted in one preschool / hospital and only 10 days menus from 4 season were analyzed (40 days in total). So the results are rather weak for generalization. Due to the reviewers comment (appearing for the second time), we will definitely come back to planned manuscript, showing the lack of seasonal differences in a large group of kindergartens.

AC: I understood the situation on the seasonal differences.

Author Response

Response to Reviewer Comments (round 2)

We greatly appreciate the time spent on reviewing our article. All the comments and suggestions were very helpful in preparing the manuscript for publication. We are sure that the quality of the manuscript has gained a lot after making the suggested changes.

Below we listed our responses to the additional comments (AC - in blue) from the round 2.

COMMENTS:

Materials and Methods:

P2: The authors should provide the materials for advertisement of this study.

Authors: Information about the program was sent via e-mail to institutions and placed in available media dedicated to institutions (e.g. educational newsletters). All these materials are only in Polish. The information included the purpose and scope of the project and a link to the website through which the institutions registered themselves to the program (http://zdrowojemy.info/placowka/zapisz-placowke-do-programu). At the moment, the link enabling registration in the program is no longer active, because the program is completed. However, the website with the program description is active and this information has been included in the manuscript text.

Now it reads: The program was nationwide; participation was voluntary and free of charge for participating institutions. The contact details for DCCs were obtained from the local relevant municipal institutions. All registred DCCs received an e-mail invitation to participate, in addition information about the program was advertised in media channels addressed to care and educational centres (press, internet portals, etc.). To participate in the program, the institutions had to register on a dedicated website (http://zdrowojemy.info/placowka/zapisz-placowke-do-programu, in Polish) until the end of the recruitment period.

AC: I thought that the contents was revised appropriately. However, typos should be corrected (e.g. “registred”, “centres”).

Authors: The typos have been corrected.

P2-3: The details of the program (both two programs) should add as the supplementary material. It because that the potential readers cannot understand the contents of the educational program through the current manuscript.

Authors: A diagram with the overall scheme of the program Healthy Eating, Healthy Growing (Figure S1) is available in supplementary materials. Information on this is included in the Materials and Methods in the section General Information and under the text of the manuscript (Supplementary Materials).

AC: I thought that the contents was revised appropriately. However, a typo should be corrected (e.g. “OUTCOM”).

Authors: The typo has been corrected.

P4: Why did the authors not assess the knowledge level of the staff members in the DCCs. It may be an important limitation of this study.

Authors: The scope of the program did not include assessing the knowledge level of employees of the institutions participating in the program. In Poland, there are numerous publications assessing the level of nutritional knowledge of care staff, both preschools and nurseries (e.g. Stankiewicz, Bogdańska: Assessment of the extent of nutritional awareness among kindergarten employees in terms of proper nutrition of preschool children. Probl Hig Epidemiol 2013, 94(3): 479-483; Chalcarz et al. Nutritional knowledge of kindergarten employees. Nowa Medycyna 7/1999, 62-67). Additionally, before the beginning of the program, an analysis of the level of nutritional knowledge was carried out among a representative group (n = 892) of the caregivers for children aged 3-6 which. Obtained results were the basis for developing the model of nutritional education used in the EHGH project (results of this study were published as a monograph, in Polish; available on-line: 148.81.185.134/exlibris/aleph/a22_1/apache_media/2U8XMSYHMNDJU4S83T33QGE4XFUL3F.pdf). The program partner (Jan Amos Comenius Children's Development Foundation) has many years of experience in this field and has been cooperating with child care institutions for many years. Both the literature data and the Foundation's experience indicate an insufficient level of nutritional knowledge and the need for staff education, which contributed to the design idea of this educational project. From a practical reason, with the assumed number of institutions (13214 nurseries and preschools employees were trained within the EHGH) verification of the knowledge of each participant would not be possible. Moreover, it seems to us that the level of initial knowledge of the staff could affect the correctness of the implementation of nutrition at baseline (before education), and not the effectiveness of education itself as educational materials were the same for all facilities and the availability of an educator (on-going support) allowed staff to ask additional questions in case of any doubts.

AC: I understood the situation, and the authors should write about the insufficient level of nutritional knowledge and the need for staff education in the Introduction and/or the Discussion section(s).

Authors: The suggested information has been added to the Introduction. The changes have been marked in purple in the manuscript text.

P6: The authors only performed univariate analysis, but they should consider factors which associate with the nutritional management (e.g. number and experiences (specialty) of staff members).

Authors: Our study group was homogeneous: only public facilities providing full board and preparing meals from scratch, which is why we conducted univariate analyzes. We did not have the consent of the bioethics commission to collect personnel data (e.g. period of employment or other professional experience). We have only obtained information about employing (or not) a dietitian for menu planning. Only in 15 institutions out of 231 a person with such specialist education was employed, which significantly reduced the validity of the analysis. We agree that having additional information about staff could enable multi-factor analysis. However, one person is always responsible for menu planning, while the number of people employed in the kitchen may vary (depending on the size of the DCC).

AC: I understood the reason why did the authors only conducted univariate analysis. However, these contexts should be included in the manuscript for the readers who does not know the situation in Poland.

Authors: The suggested information has been added to the manuscript and the changes have been marked in purple.